# Evaluation of Quinazolin-2,4-Dione Derivatives as Promising Antibacterial Agents: Synthesis, In Vitro, In Silico ADMET and Molecular Docking Approaches

**DOI:** 10.3390/molecules29235529

**Published:** 2024-11-22

**Authors:** Aboubakr H. Abdelmonsef, Mohamed El-Naggar, Amal O. A. Ibrahim, Asmaa S. Abdelgeliel, Ihsan A. Shehadi, Ahmed M. Mosallam, Ahmed Khodairy

**Affiliations:** 1Department of Chemistry, Faculty of Science, South Valley University, Qena 83523, Egypt; amalothman333@sci.svu.edu.eg (A.O.A.I.); ahmedmosallam522@gmai.com (A.M.M.); 2Pure and Applied Chemistry Group, Chemistry Department, College of Sciences, University of Sharjah, Sharjah 27272, United Arab Emirates; melnagrr@sharjah.ac.ae (M.E.-N.); ishehadi@sharjah.ac.ae (I.A.S.); 3Department of Botany and Microbiology, Faculty of Science, South Valley University, Qena 83523, Egypt; asmaa.elgafari@sci.svu.edu.eg; 4Department of Chemistry, Faculty of Science, Sohag University, Sohag 82524, Egypt; ahmed.mahmoud3@science.sohag.edu.eg

**Keywords:** quinazolin-2,4-dione, eight-membered nitrogen-heterocycles, dithiolan-4-one, phenyl-thiazolidin-4-one, antibacterial assessment, molecular docking

## Abstract

A series of new quinazolin-2,4-dione derivatives incorporating amide/eight-membered nitrogen-heterocycles **2a**–**c**, in addition, acylthiourea/amide/dithiolan-4-one and/or phenylthiazolidin-4-one **3a**–**d** and **4a**–**d**. The starting compound **1** was prepared by reaction of 4-(2,4-dioxo-1,4-dihydro-2*H*-quinazolin-3-yl)-benzoyl chloride with ammonium thiocyanate and cyanoacetic acid hydrazide. The reaction of **1** with strong electrophiles, namely, *o*-aminophenol, *o*-amino thiophenol, and/or *o*-phenylene diamine, resulted in corresponding quinazolin-2,4-dione derivatives incorporating eight-membered nitrogen-heterocycles **2a**–**d**. Compounds **3a**–**d** and **4a**–**d** were synthesized in good-to-excellent yield through a one-pot multi-component reaction (MCR) of **1** with carbon disulfide and/or phenyl isocyanate under mild alkaline conditions, followed by ethyl chloroacetate, ethyl iodide, methyl iodide, and/or concentrated HCl, respectively. The obtained products were physicochemically characterized by melting points, elemental analysis, and spectroscopic techniques, such as FT-IR, ^1^H-NMR, ^13^C-NMR, and MS. The antibacterial efficacy of the obtained eleven molecules was examined in vitro against two Gram-positive bacterial strains (*Staphylococcus aureus* and *Staphylococcus haemolyticus*). Furthermore, Computer-Aided Drug Design (CADD) was performed on the synthesized derivatives, standard drug (Methotrexate), and reported antibacterial drug with the target enzymes of bacterial strains (*S. aureus* and *S. haemolyticus*) to explain their binding mode of actions. Notably, our findings highlight compounds **2b** and **2c** as showing both the best antibacterial activity and docking scores against the targets. Finally, according to ADMET predictions, compounds **2b** and **2c** possessed acceptable pharmacokinetics properties and drug-likeness properties.

## 1. Introduction

Nitrogen-heterocyclic compounds are of particular interest by virtue of their biological and pharmacological activity [1,2,3,4,5]. Quiet recently, compounds incorporating a quinazolin-2,4-dione moiety represented an inexhaustible inspiration for the design and development of novel semisynthetic or synthetic agents with a broad spectrum of bioactivities [6,7]. Quinazolin-2,4-diones stand out as promising candidates in pharmacology, having several biological activities, including anticancer [8], antibacterial [9], anti-malarial [10], and anti-inflammatory [8].

Eight-membered nitrogen-heterocycles such as azocine are considered privileged structures found in a variety of natural products and bioactive molecules [11,12,13,14]. They play a fundamental role in medicinal and pharmaceutical chemistry. They serve as a key scaffold for the design and development of various inhibitors, including broad-spectrum antibacterial drug candidates [15,16]. For example, AZOCIN-500^®^ (Azithromycin) is an antibiotic that is utilized in the treatment of bacterial infections and typhoid fever. AZOCIN-500^®^ stops bacterial growth and infection spread [12,17].

Recent studies on derivatives incorporating amide and acylthiourea moieties exhibited a broad spectrum of biological applications, e.g., antibacterial, antiviral, and antioxidant activity [18,19,20,21,22].

Thiazolidin-4-one is considered an essential heterocyclic scaffold in medicinal chemistry. Moreover, it has a broad range of biological activities, including antibacterial, anticancer, and anti-inflammatory [23,24,25], Figure 1.

In addition, the molecular hybridization approach is responsible for good antibacterial activity [26].

Inspired by the data collected, as well as in continuation of our efforts to synthesize new and promising antibacterial inhibitors [19,27]. Herein, a new series of eleven compounds with various bioactive moieties such as quinazolin-2,4-dione, amide, eight-membered nitrogen-heterocycles, acylthiourea, dithiolan-4-one and/or phenyl-thiazolidin-4-one were synthesized. Virtual screening on diverse quinazolin-2,4-dione derivatives and standard drugs to unveil their inhibition potential against the target enzymes was performed. Furthermore, biological evaluations of the new quinazolin-2,4-dione derivatives were performed against two Gram-positive bacterial strains, namely, *Staphylococcus aureus* and *Staphylococcus haemolyticus*. Finally, the ADME/Tox and drug-likeness properties of the best-docked compounds and methotrexate were checked using AdmetSAR, Mol inspiration, and SwissADME web servers.

## 2. Results and Discussion

### 2.1. Chemistry

Compound **1** was reported earlier in our previous study [18]. Compounds **2a**–**c**, quinazolin-2,4-diones attached to eight-membered nitrogen-heterocycles, such as oxa/thia/tri/tetr-azocine, were synthesized via the reaction of **1** with strong electrophiles, namely, *o*-aminophenol, *o*-aminothiophenol, and *o*-phenylene diamine, respectively (Figure 1). The chemical structures of the new compounds **2a**–**c** were elucidated based on spectroscopic data. For instance, the FT-IR spectrum of **2a** showed bands at 3180, 1709, 1660, and 1609 cm^−1^ attributed to NH, C=O, and C=N groups, respectively. A band at 2265 cm^−1^ was assignable to a C≡N group, indicating that the nucleophilic attack did not occur at the C≡N group. The ^1^H-NMR spectrum of **2a** showed signals attributed to NH, CH_2_N, and aromatic protons at 11.62, 4.02, and 7.74–8.04 ppm. Further evidence was gained from the mass spectrum; it showed the correct molecular ion peak at *m*/*z* 479 beside some other important peaks.

On the other hand, compounds with an activated methylene group react as carbanions in the presence of a base with the electrophilic carbon disulfide to yield dithiocarboxylates. This can be converted to ketene dithioacetals on treatment with an excess of the alkylating reagent [28]. Cyclization of the intermediate (A1) with ethyl chloroacetate afforded **3a**. The reaction proceeded via nucleophilic addition of the carbanions on CS_2_ to form the potassium salt intermediates (A1), followed by in situ cyclization through an SN^2^ mechanism to yield the cyclic compound **3a** (Figure 2). Stirring of **1** with carbon disulfide in the presence of KOH in DMF followed by the addition in situ of ethyl iodide or methyl iodide or concentrated HCl afforded compounds **3b**–**d** via intermediate (A1) (Figure 2). The structures of compounds **3a**–**d** were established by means of analytical and spectral data. The IR spectra of compounds **3a**–**d** showed bands characteristic for NH, CN, C=O, and C=S groups in the range 3187–3110, 2205–2250, and 1722–1662 cm^−1^, respectively. The ^1^H-NMR spectra are in good agreement with the suggested structures. They are devoid of a signal corresponding to CH_2_CN protons. However, they displayed signals related to two CH_3_ and two CH_3_S protons for compounds **3b** and **3c** at δ 1.36 ppm as triplet signals and 2.68 ppm as singlet signals, respectively. Two SCH_2_ and SCH_2_CO protons for compounds **3b** and **3a** appeared at δ 2.83 ppm as a quartet signal and δ 4.00 ppm as a singlet signal. The ^13^C-NMR spectra of compounds **3a**–**c** exhibited signals at δ 34.55, 19.10, 24.35, and 14.44 ppm, respectively, indicating the presence of SCH_2_CO, CH_3_, CH_2_S, and SCH_3_. Further evidence was gained from the mass spectra, as they showed the correct molecular ion peaks for compounds **3a**–**d** at *m*/*z* 538, 554, 526, and 498, respectively.

Furthermore, the reaction of compound **1** with phenyl isothiocyanate in the presence of KOH yielded the potassium salt intermediate (A2). Cyclization of (A2) in situ with ethyl chloroacetate furnished compound **4a**. Also, followed by the addition in situ of ethyl iodide, methyl iodide, or concentrated HCl afforded compounds **4b**–**d** via the molecule intermediate (A2) (Figure 1). The structures of compounds **4a**–**d** were elucidated on the basis of the elemental analysis and spectral data. The FT-IR spectra of compounds **4a**–**d** displayed absorption bands corresponding to NH, CN, C=O, and C=S groups at 3182, 3333, 3211, 2203, 2202, 1673, and 1714 cm^−1^, respectively. Furthermore, the ^1^H-NMR spectrum of compound **4a** showed characteristic signals for four NHs, aromatic protons, and CH_2_ protons at 11.65, 11.61, 10.34, 9.61, 7.24–7.79, and 4.00 ppm, respectively. An extended analysis was performed using the mass spectrum, the recorded mass *m*/*z* at (597 [M]^+^) corresponding to the calculated molecular formula. The spectral data of all newly prepared quinazolin-2,4-dione derivatives are declared in Appendix A.

### 2.2. Biological Studies

In the present study, all the synthesized compounds were screened for their in vitro antibacterial activity using MIC and MBC assays. Table 1 declares the minimum inhibitory concentration (MIC) and minimum bactericidal concentration (MBC) values of compounds tested against two Gram-positive bacterial strains: *Staphylococcus aureus* and *Staphylococcus haemolyticus*. The MIC values ranged from 10 to 26 (mg/mL), indicating varying levels of antibacterial activity.

Most of the tested compounds exhibited significant antibacterial properties, with marked differences observed between the MIC and MBC values. The lowest MIC values were recorded for compounds **2b** and **2c**, which both exhibited lower MICs, as well as **2b** with 10 mg/mL against *S. haemolyticus*. Additionally, compound **2c** showed MIC of 11 mg/mL against *S. aureus*, followed by **3c** with 12 mg/mL, and **2a** with 13 mg/mL against *S. haemolyticus*. Thus, both the **2b** and **2c** compounds have promising antibacterial activity against the two tested G+ve bacteria. Herein, a structure–activity relationship (SAR) study is reported, which focuses on the presence of –CH_2_CN (cyanomethyl), amide, and/or thia/tri/tetr-azocine moieties, respectively. Gui Z et al. 2013 [29] reported that the function group of azocine in the antibiotic Azithromycin reduces the production of α-hemolysin and biofilm formation in *S. aureus*.

Regarding bactericidal activity, the lowest MBC value was found for **3a** at 10 mg/mL against *S. aureus*, followed closely by **4d** with an MBC of 11 mg/mL. Both compounds **2b** and **4c** demonstrated MBC values of 13 mg/mL against *S. haemolyticus* and *S. aureus*, respectively. Conversely, the highest MIC and MBC values were observed for **3b** against both bacterial strains, indicating reduced efficacy. The main backbone of the tested compounds is a quinazoline-2,4-dione moiety, which was previously described as an inhibitor for enzymes of dihydrofolate reductase and purine synthesis in microorganisms [30].

Overall, compounds **2b** and **2c** exhibited the highest antibacterial activity (due to the molecular hybridization between quinazolin-2,4-dione scaffold and/or thia/tri/tetr-azocine moieties); all showed the lowest MIC values of 10 mg/mL, making them the most promising antibacterial agents in this study. The analysis of the MBC/MIC ratios is depicted in Figure 2, illustrating that most compounds had ratios ≤2, suggesting a strong bactericidal effect.

### 2.3. In Silico Studies and ADMET Analysis

In this study, a set of quinazolin-2,4-dione derivatives was screened by CADD to identify compounds showing potent enzyme activity and acceptable pharmacokinetic properties. Dihydrofolate reductase DHFR is considered an essential enzyme for thymidylate and purine synthesis in microorganisms [31]. In addition, the literature suggested that eukaryotic initiation factor 2 α (eIF2α) signaling may be active during bacterial infections [32]. Therefore, dihydrofolate reductase and eukaryotic initiation factor 2 α were selected as promising targets for the identification of new antibacterial inhibitors. Herein, in silico molecular docking studies were performed for a set of quinazolin-2,4-diones against the target enzymes of bacterial strains, dihydrofolate reductase (PDB ID: 2W9S), and eukaryotic initiation factor 2 α (eIF2α) (PDB ID: 1Q46) utilizing a PyRx-virtual screening tool [33]. For the standard drug (Methotrexate) and the reported antibacterial drug, the docking study was also performed in order to map important interactions with the active site of the targets. The results obtained from the docking study are depicted in Table 2. Figure 3 and Figure 4 exhibited 2D and 3D interactions between the best-docked compounds and standard drugs with the target enzymes.

In the case of dihydrofolate reductase, compound **2b** (with thia/triazocine moiety) exhibited the best binding affinity, −11.7 kcal/mol, and docked to the target enzyme through one H-bond, two arene-arene, and one arene-sigma interaction with the residues ASN18, PHE92, and LEU20, while compound **2c** (with tetrazocine moiety) showed binding energy of −11.6 kcal/mol and docked to the target through one H-bond and two arene-arene interactions with the residues ASN18 and PHE9.

In the case of eukaryotic initiation factor 2 α, compound **2b** (−9.6 kcal/mol) docked to the target through two H-bonds and aren–cation interactions with the residues TYR171, TYR141, and ARG175. On the other hand, compound **2c** (−9.5 kcal/mol) docked to the target through one H-bond and two arene–arene interactions with the residues TYR141 and ARG175, respectively.

For methotrexate (−9.3 kcal/mol), six H-bonds with dihydrofolate reductase through ARG44, LYS45, LEU62, and ASN64. In addition, it docked with eukaryotic initiation factor 2 α (−7.1 kcal/mol) through five H-bonds and one arene–cation interaction.

For reported antibacterial drug [34], it docked with dihydrofolate reductase (−9.3 kcal/mol) through one H-bond with PHE92 at 2.5 Å. Additionally, it docked with eukaryotic initiation factor 2 α (−7.0 kcal/mol) through one H-bond with the residue ARG175 at 2.17 Å.

The 3D interactions of the other docked compounds toward the target enzymes are represented in Appendix A.

By comparing the experimental antibacterial activity of the compounds reported in this study to their structures, the following structure–activity relationship (SAR) was postulated:

Compounds **2b** and **2c** exhibited the highest antibacterial activity, which may be due to the presence of –CH_2_CN, amide, and/or thia/tri/tetr-azocine moieties, respectively. In addition, it was reported that the -C=N- bond is utilized in the design of antibacterial agents [35]. Further, the molecular hybridization between the quinazoline-2,4-dione scaffold and/or the thia/tri/tetr-azocine moieties is responsible for good antibacterial activity [26].

Table 3 declares the ADMET properties of the best-docked molecules, standard drugs, and reported antibacterial drugs. Their molecular weights are below 500 g/mol, indicating good absorption. Consequently, they have satisfied the Lipinski rule without any violation. They have rotatable bonds within the allowed range (<8 bond) that enhance their flexibility. In addition, they have acceptable HBA and HBD. In conclusion, compounds **2b** and **2c** are predicted to have acceptable bioavailability.

## 3. Experimental

### 3.1. Organic Synthesis

An electrothermal melting apparatus was used to measure the melting points, which were uncorrected. All chemical reactions were observed on a silica gel GF254 plate with thin-layer chromatography (TLC). FT-IR spectra υ/cm^−1^ (KBr) were recorded on a Shimadzu 8101 PC spectrometer from South Valley University. The ^1^H- and ^13^C-NMR spectra were run on a Varian Mercury spectrophotometer at 400 and 100 MHz, respectively, using tetramethylsilane TMS as an internal standard and DMSO-*d*_6_ as a solvent. Electron impact mass spectra were obtained at 70 eV using a GCMS-QP 1000 EX spectrometer. Elemental analyses were carried out at the microanalytical center at Cairo University.


**Synthesis of *N*-[*N*′-(2-cyano-acetyl)-hydrazino-carbo-thioyl]-4-(2,4-dioxo-1,4-dihydro-2H-quinazolin-3-yl)-benzamide 1**


The compound was described earlier by our group members [18].


**General procedures for synthesis of oxa/thia/triazocinyl/tetrazocinyl quinazolin-2,4-diones 2a–c**


To a solution of compound **1** (0.003 mol, 1.5 g) in DMF (30 mL), *o*-aminophenol and/or *o*-aminothiophenol and/or *o*-phenylene diamine (0.003 mol) was added to the mixture. The reaction mixture was refluxed for 10 h. The separated solid was filtrated off, dried, and recrystallized to afford compounds **2a**–**c**, respectively.


**Synthesis of *N*-2-(cyanomethyl)-4H-benzo-[g]-[1,3,4,6]-oxatriazocin-5-yl)-4-(2,4-dioxo-1,4-dihydroquinazolin-3(2H)-yl)-benzamide 2a**


Dark brown crystals. Yield 62%; MP 226–228 °C. FT-IR (KBr, υ, cm^−1^) = 3180 (NH’s), 2265 (CN), 1709, 1660 (C=O’s), 1608 (C=N). ^1^H-NMR (DMSO-*d*_6_, 400 MHz): δ (ppm) = 11.62 (s, 1H, NH), 7.74–8.04 (m, 14H, Ar-H+2NH), 4.02 (s, 2H, CH_2_). ^13^C-NMR (DMSO-*d*_6_, 100 MHz): δ (ppm)= 19.13, 114.68, 114.78, 114,79, 114.80, 115.78, 115.89, 116.13, 123.09, 128.07, 128.47, 129.82, 135.80, 140.34, 142.49, 143.51, 150.56, 156.15, 157.56, 159.09, 160.00, 160.49, 161.07, 162.47, 163.18. MS (EI): *m*/*z* (%) = 479 [M]^+^. *Anal. Calcd* for C_25_H_17_N_7_O_4_ (Mol. Wt.: 479): C, 62.63; H, 3.57; N, 20.45%. Found C, 62.75; H, 3.69; N, 20.33%.


**Synthesis of *N*-2-(cyanomethyl)-4H-benzo-[g]-[1,3,4,6]-thiatriazocin-5-yl)-4-(2,4-dioxo-1,4-dihydroquinazolin-3(2H)-yl)-benzamide 2b**


Dark green crystals. Yield 65%; MP > 300 °C. FT-IR (KBr, υ, cm^−1^) = 3195 (NH’s), 2053 (CN), 1710, 1671 (C=O’s), 1612 (C=N). ^1^H-NMR (DMSO-*d*_6_, 400 MHz): δ (ppm) = 11.61 (s, 1H, NH), 7.14–8.22 (m, 14H, Ar-H+2NH), 4.17 (s, 2H, CH_2_). ^13^C-NMR (DMSO-*d*_6_, 100 MHz): δ (ppm) = 20.20, 114.81, 115,26, 115.79, 116.52, 116.92, 122.95, 123.09, 123.51, 126.19, 127.26, 128.09, 128.14, 130.79, 131.61, 133.11, 135.18, 135.81, 135.89, 138.93, 140.34, 150.23, 150.52, 154.09, 162.56, 167.11. MS (EI): *m*/*z* (%) = 495 [M]^+^. *Anal. Calcd* for C_25_H_17_N_7_O_3_S (Mol. Wt.: 495): C, 60.60; H, 3.46; N, 19.79; S, 6.47%. Found C, 60.72; H, 3.58; N, 19.68; S, 6.58%.


**Synthesis of *N*-5-(cyanomethyl)-3,6-dihydrobenzo[e]-[1,2,4,7]-tetrazocin-2-yl)-4-(2,4-dioxo-1,4-dihydroquinazolin-3(2H)-yl) benzamide 2c**


Pale brown crystals. Yield 70%; MP 280–282 °C. FT-IR (KBr, υ, cm^−1^) = 3190 (NH’s), 2275 (CN), 1723, 1665 (C=O’s), 1610 (C=N). ^1^H-NMR (DMSO *d*_6_, 400 MHz): δ (ppm) = 11.57 (s, 1H, NH), 7.19–8.23 (m, 14H, Ar-H+2NH), 3.96 (s, 2H, CH_2_). ^13^C-NMR (DMSO-*d*_6_, 100 MHz): δ (ppm) = δ 21.51, 114.63, 115,78, 115.79, 122.13, 123.08, 128.02, 128.06, 128.07, 129.33, 129.66, 130.18, 135.78, 135.79, 140.33, 140.34, 150.51, 150.79, 151.51, 162.63, 162.64, 162.66, 172.33, 174.50. MS (EI): *m*/*z* (%) = 478 [M]^+^. *Anal. Calcd* for C_25_H_18_N_8_O_3_ (Mol. Wt.: 478): C, 62.76; H, 3.79; N, 23.42%. Found C, 62.87; H, 3.91; N, 23.32%.


**General procedures for the synthesis of compounds 3a–d**


To a stirred suspension of finely powdered potassium hydroxide (0.002 mol, 1.12 g) in dry DMF (20 mL), compound 1 (0.002 mol, 1 g) was added. The resulting mixture was cooled at 10 °C in an ice bath, and then carbon disulfide (0.50 mL, 0.002 mol) was added slowly over the course of 10 min. After complete addition, stirring of the reaction mixture was continued for an additional 4 h. Then, ethyl chloroacetate, ethyl iodide, methyl iodide, or concentrated HCl (0.002 mol) was added to the mixture while cooling and stirring for 20 h. The mixture was then poured onto crushed ice; the resulting precipitate was filtrated off, dried, and recrystallized from the proper solvent to give compounds **3a**–**d**, respectively.


**Synthesis of *N*-(2-(2-cyano-2-(4-oxo-1,3-dithiolan-2-ylidene)acetyl)hydrazine-1-carbono thioyl)-4-(2,4-dioxo-1,4-dihydroquinazolin-3(2H)-yl)benzamide 3a**


Orange crystals. Yield: 63%. MP > 300 °C. FT-IR (KBr, υ, cm^−1^) = 3187, 2995 (NH’s), 2205 (CN), 1718, 1662 (C=O’s), 1271 (C=S). ^1^H-NMR (DMSO-*d*_6_, 400 MHz): δ (ppm) = 11.66 (s, 1H, NH), 11.64 (s, 1H, NH), 10.72 (s, 1H, NH), 7.26–8.06 (m, 8H, Ar-H), 4.00 (s, 2H, CH_2_). ^13^C-NMR (DMSO-*d*_6_, 400 MHz): δ (ppm) = 34.55, 94.94, 114.74, 115.80, 115.82, 116.17, 123.09, 128.03, 128.55, 129.51, 129.90, 129.95, 130.29, 132.43, 135.80, 139.47, 140.34, 150.43, 150.60, 162.33, 162.60, 165.51. MS (El): *m*/*z* (%) = 538 [M]^+^. *Anal. Calcd* for C_22_H_14_N_6_O_5_S_3_ (Mol. Wt.: 538): C, 49.06; H, 2.62; N, 15.60; S, 17.86%, found: C, 49.21; H, 2.75; N, 15.49; S, 17.98%. 


**Synthesis of *N*-(2-(2-cyano-3,3-bis-(ethylthio)-acryloyl)-hydrazine-1-carbonothioyl)-4-(2,4-dioxo-1,4-dihydroquinazol-in-3-(2H)-yl)benzamide **
**3**
**b**


Yellowish brown crystals. Yield: 65%. MP 140–142 °C. FT-IR (KBr, υ, cm^−1^) = 3135 (NH), 2235 (CN), 1722, 1671 (C=O’s), 1348 (C=S). ^1^H-NMR (DMSO-*d*_6_, 400 MHz): δ (ppm) = 11.69 (s, 1H, NH), 10.69 (s, 1H, NH), 10.55 (s, 1H, NH), 8.87 (s, 1H, NH), 7.23–8.10 (m, 8H, Ar-H), 2.81–2.85 (q, 4H, 2CH_2_), 1.34–1.38 (t, 6H, 2CH_3_). ^13^C-NMR (DMSO-*d*_6_, 100 MHz): δ (ppm) = 19.10, 24.35, 102.01, 114.76, 115,79, 116.13, 123.10, 128.06, 128.54, 129.91, 132.42, 135.81, 139.46, 140.32, 150.46, 161.03, 162.33, 162.59, 165.36, 165.58, 171.88, 184.20. MS (El): *m*/*z* (%) = 554 [M]^+^. *Anal. Calcd* for C_24_H_22_N_6_O_4_S_3_ (Mol. Wt.: 554): C, 51.97; H, 4.00; N, 15.15; S, 17.34%, found: C, 52.05; H, 4.13; N, 15.02; S, 17.43%.


**Synthesis of *N*-(2-(2-cyano-3,3-bis(methylthio)-acryloyl)-hydrazine-1-carbonothioyl)-4-(2,4-dioxo-1,4-dihydroquinazolin-3(2H)-yl)benzamide 3**
**c**


Pale yellow powder. Yield: 60%. MP 120–122 °C. FT-IR (KBr, υ, cm^−1^) = 3110, 2925 (NH’s), 2250 (CN), 1719, 1670 (C=O’s), 1271 (C=S). ^1^H-NMR (DMSO-*d*_6_, 400 MHz): δ (ppm) = 11.64 (s, 1H, NH), 10.67 (s, 1H, NH), 10.60 (s, 1H, NH), 10.45 (s, 1H, NH), 7.24–7.99 (m, 8H, Ar-H), 2.68 (s, 6H, 2CH_3_). ^13^C-NMR (DMSO-*d*_6_, 100 MHz): δ (ppm) = 14.44, 114.76, 115,80, 123.09, 128.05, 128.33, 128.48, 129.52, 129.77, 129.93, 135.80, 138.55, 139.41, 140.34, 150.49, 162.60, 162.59, 165.67, 165.76, 168.80, 182.49. MS (El): *m*/*z* (%) = 526 [M]^+^. *Anal. Calcd* for C_24_H_18_N_6_O_4_S_3_ (Mol. Wt.: 526): C, 50.18; H, 3.45; N, 15.96; S, 18.26%, found: C, 50.30; H, 3.59; N, 15.84; S, 18.39%.


**Synthesis of N-(2-(2-cyano-3,3-dimercaptoacryloyl)-hydrazine-1-carbono-thioyl)-4-(2,4-dioxo-1,4-dihydroquinazolin-3(2H)-yl)benzamide 3d**


Yellow crystals. Yield: 70%. MP 208–210 °C. FT-IR (KBr, υ, cm^−1^) = 3489 (SH), 3200, 3135 (NH’s), 2230 (CN), 1718, 1668 (C=O’s), 1272 (C=S). ^1^H-NMR (DMSO-*d*_6_, 400 MHz): δ (ppm) = 11.68 (s, 1H, NH), 10.69 (s, 1H, NH), 10.53 (s, 1H, NH), 8.81 (s, 1H, NH), 7.23–7.98 (m, 8H, Ar-H), 1.24 (s, 2H, SH). ^13^C-NMR (DMSO-*d*_6_, 100 MHz): δ (ppm) = 101.04, 114.73, 114.74, 115.82, 115.83, 123.09, 128.04, 128.55, 129.90, 129.91, 132.56, 135.81, 140.34, 150.43, 150.44, 162.60, 162.74, 162.75, 165.75, 181.04. MS (El): *m*/*z* (%) = 498 [M]^+^. *Anal. Calcd* for C_20_H_14_N_6_O_4_S_3_ (Mol. Wt.: 498): C, 48.18; H, 2.83; N, 16.86; S, 19.29%, found: C, 48.30; H, 2.95; N, 16.74; S, 19.32%.


**General procedures for the synthesis of compounds 4a–d**


To a dissolved compound **1** (0.003 mol, 1.5 g) in (DMF) (20 mL), potassium hydroxide (0.003 mol, 0.2 g) was added. The mixture was stirred at RT until the complete dissolution of potassium hydroxide, and then phenyl isothiocyanate (0.003 mol, 0.47 g) was added after completing the stirring for 5 h. After that, ethyl chloroacetate, ethyl iodide, methyl iodide, or concentrated HCl (0.003 mol) were added with stirring overnight. Then, it was quenched into water and acidified with 10% hydrochloric acid, and the obtained products **4a**–**d** were collected by filtration and recrystallized, respectively.


**Synthesis of *N*-(2-(2-cyano-2-(4-oxo-3-phenylthiazolidin-2-ylidene)-acetyl)hydrazine-1-carbonothioyl)-4-(2,4-dioxo-1,4-dihydroquinazolin-3(2H)-yl)benzamide 4a**


Yellow crystals. Yield: 57%. MP 90–92 °C. FT-IR (KBr, υ, cm^−1^) = 3205, 3058 (NH’s), 2088 (CN), 1719, 1667 (C=O’s), 1269 (C=S). ^1^H-NMR (DMSO-*d*_6_, 400 MHz): δ (ppm) = 11.65 (s, 1H, NH), 11.61 (s, 1H, NH), 10.34 (s, 1H, NH), 9.61 (s, 1H, NH), 7.24–7.79 (m, 13H, Ar-H), 4.00 (s, 2H, CH_2_). ^13^C-NMR (DMSO-*d*_6_, 100 MHz): δ (ppm) = 32.99, 68.59, 114.57, 114.77, 115.79, 115.80, 123.09, 126.50, 128.06, 128.21, 128.76, 128.90, 129.26, 129.86, 129.96, 135.81, 135.82, 140.33, 142.99, 147.80, 150.43, 150.46, 158.79, 162.59, 165.26, 175.57, 184.20. MS (El): *m*/*z* (%) = 597 [M]^+^. *Anal. Calcd* for C_28_H_19_N_7_O_5_S_2_ (Mol. Wt.: 597): C, 56.27; H, 3.20; N, 16.41; S, 10.73%, found: C, 56.40; H, 3.33; N, 16.38; S, 10.85%.


**Synthesis of N-(2-(2-cyano-3-(ethylthio)-3-(phenylamino)-acryloyl)hydrazine-1-carbono thioyl)-4-(2,4-dioxo-1,4-dihydroquinazolin-3(2H)-yl)benzamide 4b**


Yellow crystals. Yield: 55%. MP 158–160 °C. FT-IR (KBr, υ, cm^−1^) = 3262, 3070 (NH’s), 2213 (CN), 1717, 1693 (C=O’s), 1258 (C=S). ^1^H-NMR (DMSO-*d*_6_, 400 MHz): δ (ppm) = 12.17 (s, 1H, NH),11.64 (s, 1H, NH), 11.60 (s, 1H, NH), 10.63 (s, 1H, NH), 9.81 (s, 1H, NH), 7.22–7.98 (m, 13H, Ar-H), 3.15–3.21 (q, 2H, CH_2_), 1.34–1.38 (t, 3H, CH_3_). ^13^C-NMR (DMSO-*d*_6_, 100 MHz): δ (ppm) = 14.74, 26.97, 115,78, 123.06, 124.12, 124.89, 126.44, 127.00 128.36, 128.49, 128.74, 128.81, 128.92, 129.10, 129.62, 129.80, 129.92, 130.40, 134.42, 135.78, 137.39, 140.26, 140.32, 150.43, 150.46, 152.64, 154.33, 162.55, 180.08. MS (El): *m*/*z* (%) = 585 [M]^+^. *Anal. Calcd* for C_28_H_23_N_7_O_4_S_2_ (Mol. Wt.: 585): C, 57.42; H, 3.96; N, 16.74; S, 10.95%, found: C, 57.55; H, 4.06; N, 16.64; S, 11.05%.


**Synthesis of *N*-(2-(2-cyano-3-(methylthio)-3-(phenylamino)-acryloyl)hydrazine-1-carbono thioyl)-4-(2,4-dioxo-1,4-dihydroquinazolin-3(2H)-yl)benzamide 4c**


Orange crystals. Yield: 65%. MP > 300 °C. FT-IR (KBr, υ, cm^−1^) = 3205, 3008 (NH’s), 2260 (CN), 1714, 1666 (C=O’s), 1269 (C=S). ^1^H-NMR (DMSO-*d*_6_, 400 MHz): δ (ppm) = 11.66 (s, 1H, NH), 11.64 (s, 1H, NH), 9.60 (s, 1H, NH), 9.59 (s, 1H, NH), 7.28–7.71 (m, 13H, Ar-H), 1.21 (s, 3H, CH_3_). MS (El): *m*/*z* (%) = 571 [M]^+^. *Anal. Calcd* for C_27_H_21_N_7_O_4_S_2_ (Mol. Wt.: 571): C, 56.73; H, 3.70; N, 17.15; S, 11.22%, found: C, 56.85; H, 3.82; N, 17.02; S, 11.34%.


**Synthesis of N-(2-(2-cyano-3-(phenylamino)-3-thioxopropanoyl)hydrazine-1-carbono thioyl)-4-(2,4-dioxo-1,4-dihydroquinazolin-3(2H)-yl)benzamide 4d**


Pale yellow crystals. Yield: 58%. MP > 300 °C. FT-IR (KBr, υ, cm^−1^) = 3262, 3069, 3006 (NH’s), 2213 (CN), 1718, 1664 (C=O’s), 1258 (C=S). ^1^H-NMR (DMSO-*d*_6_, 400 MHz): δ (ppm) = 9.02 (s, 1H, NH), 7.10–7.31 (m, 14H, Ar-H+CH). MS (El): *m*/*z* (%) = 557 [M]^+^. *Anal. Calcd* for C_26_H_19_N_7_O_4_S_2_ (Mol. Wt.: 557): C, 56.01; H, 3.43; N, 17.58; S, 11.50%, found: C, 56.23; H, 3.55; N, 17.46; S, 11.62%.

### 3.2. Antibacterial Susceptibility Testing

#### 3.2.1. Bacterial Strains and Culture Conditions

The human pathogenic Gram-positive bacteria *Staphylococcus aureus* and *Staphylococcus haemolyticus* were used in this study. Bacterial strains were kindly obtained from the Faculty of Science—Botany and Microbiology Department—Bacteriology Laboratory. Bacterial strains were maintained on Tryptic Soy Agar (TSA) slants and incubated at 37 °C for 24–48 h. The inocula were spread over (TSA) plates prior to the antimicrobial activity tests.

#### 3.2.2. Determination of Minimum Inhibitory Concentration (MIC) by INT Assay

The antibacterial activities of the compounds were assessed using MIC and MBC assays. The MIC, defined as the lowest concentration that inhibits visible bacterial growth after overnight incubation, was determined using the INT assay. Sterile 96-well microtiter plates were employed, with each well containing a 100 µL bacterial suspension adjusted to a 0.001 = OD_595_ [36] and 10 µL serial dilutions of the chemical compounds.

The plates were incubated at 37 °C for 24 h, followed by the addition of INT (p-iodonitrotetrazolium violet) to assess bacterial growth. A total of 60 µL of INT (p-iodonitrotetrazolium violet, 0.2 mg mL^−1^) was added to microplate wells and re-incubated at 37 °C for 2 h [37]. The MIC in the INT assay was defined as the lowest concentration of chemical substances that prevented color change, indicating bacterial growth inhibition, as described earlier [36]. All the experiments were performed in eight replicates represented by one column in the 96-well plates.

#### 3.2.3. Determination of Minimum Bactericidal Concentration (MBC)

The MBC, which represents the lowest concentration that completely eliminates the bacteria, was determined by sub-culturing 20 µL of the suspension from MIC wells onto sterile tryptic soya agar plates [38]. The MBC was determined by transferring 20 microliters of suspension from each well of overnight incubated MIC plates and inoculated on sterile tryptic soya agar in fresh plates with continuous shaking with sterilized glass beads (0.4 mm) and incubated at 37 °C for 24 h. The MBC-causing bactericidal effect was identified on the basis of colony absence on the agar plates [39].

### 3.3. In Silico Studies

The molecular docking studies were performed for a set of quinazolin-2,4-diones and a standard drug and reported antibacterial drug against the targets dihydrofolate reductase (PDB ID: 2W9S) and eukaryotic initiation factor 2 α (eIF2α) (PDB ID: 1Q46) utilizing the PyRx-virtual screening tool [33]. The crystal structures of the target enzymes were obtained from the RCSB Protein Data Bank web server. Subsequently, the target files were optimized by removing the ligands and water molecules. Their energies were minimized using CHARMm Force Field [40] in Discovery Studio 3.5 Visualizer. In addition, the prepared molecules, methotrexate, and the reported antibacterial drug were sketched in cdx format (2D structures) using ChemDraw Ultra 8.0 and then were converted to sdf files (3D structures) by using Open Babel GUI 2.4.1 tool [41]. The energy of the synthesized molecules was minimized in the PyRx tool with default parameters (UFF force field) [42] and then docked flexibly to the targets. The visualizations of docking results were performed using Discovery Studio 3.5. Finally, the ADMET properties of the best-docked molecules and standard drugs were investigated using the AdmetSAR and SwissADME web servers.

## 4. Conclusions

A series of new quinazolin-2,4-dione derivatives were prepared with from good to excellent yields. Chemical structures and purity were proven from spectral data and elemental analysis. Antibacterial efficacies for new derivatives were assessed in vitro against two Gram-positive bacterial strains (*S. aureus* and *S. haemolyticus*). The molecules **2b** and **2c** exhibited good antibacterial activity, which may be due to the presence of quinazolin-2,4-dione, –CH_2_CN, amide, and/or thia/tri/tetr-azocine moieties, respectively.

Additionally, computer-aided drug design (CADD) was carried out to screen the new quinazolin-2,4-dione derivatives, standard drug, and reported antibacterial drug against the target enzymes to establish the mechanism by which the molecules inhibit the growth of *S. aureus* and *S. haemolyticus*. It is noteworthy that the data obtained from the in silico docking study were in excellent agreement with the in vitro antibacterial results.

## Data Availability

All data generated or analyzed during this study are included in the Appendix A.

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
