# Peer review of "Evaluation of Quinazolin-2,4-Dione Derivatives as Promising Antibacterial Agents: Synthesis, In Vitro, In Silico ADMET and Molecular Docking Approaches"

_molecules, 2024, doi:10.3390/molecules29235529_

Round 1
Reviewer 1 Report
Comments and Suggestions for Authors
The manuscript "Evaluation of quinazolin-2,4-dione Derivatives as Promising 2 Antibacterial Agents: Synthesis, In vitro, In silico ADMET and 3 Molecular Docking Approaches" describes 11 novel quinazolin-2,4-dione derivatives, their physicochemical characterization and antibacterial efficacy in vitro against two gram-positive bacterial. The goals and motivation of such a study are relatively straightforward. After considering the issues outlined below, the rewritten paper can contribute to Molecules.
Additional comments that need to be addressed:
1) Figure 1 is redundant.
2) In Chapter 2.1, the details of FT IR and C NMR are unnecessary; characteristic signals are sufficient.
3) The results are very poorly described and unexplained. I feel like I am reading data taken from tables. Please explain, for example, why the lowest MIC value was recorded for derivatives 4c, 3a, and 2b. etc.
4) The biological and in silico studies and ADMET analysis should be compared carefully with data from the literature for structurally similar compounds.
Comments on the Quality of English Language
Although the manuscript is written well enough in English to be understood, the English language and style (grammar) must be carefully revised throughout the whole manuscript.
Author Response
Comments and Responses # molecules 3251295 (Reviewer 1)
The manuscript "Evaluation of quinazolin-2,4-dione Derivatives as Promising 2 Antibacterial Agents: Synthesis, In vitro, In silico ADMET and 3 Molecular Docking Approaches" describes 11 novel quinazolin-2,4-dione derivatives, their physicochemical characterization and antibacterial efficacy in vitro against two gram-positive bacterial. The goals and motivation of such a study are relatively straightforward. After considering the issues outlined below, the rewritten paper can contribute to Molecules.
We are thankful to the reviewers for their valuable comments on our manuscript (Manuscript Number: # molecules 3251295) entitled “Evaluation of quinazolin-2,4-dione Derivatives as Promising Antibacterial Agents: Synthesis, In vitro, In silico ADMET and Molecular Docking Approaches”. The constructive suggestions and comments of the reviewers have greatly helped us in improving our manuscript. We have revised the manuscript in the light of comments of the reviewers, and the corrections have been highlighted in the revised manuscript. The point wise reply to the comments of the learned reviewer 1 are provided below.
Additional comments that need to be addressed:
- Figure 1 is redundant.
Response
The necessary corrections are included in the revised manuscript.
- In Chapter 2.1, the details of FT IR and C NMR are unnecessary; characteristic signals are sufficient.
Response
According to the reviewer's suggestion, the necessary corrections are included in the revised manuscript.
- The results are very poorly described and unexplained. I feel like I am reading data taken from tables. Please explain, for example, why the lowest MIC value was recorded for derivatives 4c, 3a, and 2b. etc.
Response
According to the reviewer suggestion, the results of biological section are improved in the revised manuscript.
- The biological and in silico studies and ADMET analysis should be compared carefully with data from the literature for structurally similar compounds.
Response
As per the reviewer's suggestion, the necessary corrections are included in the revised manuscript.
We understand the importance of standard drug (Methotrexate) in biological study, but unfortunately it is not available to do this study, in addition to unavailability of the strains.

Reviewer 2 Report
Comments and Suggestions for Authors
In this manuscript, A series of new eleven quinazolin-2,4-dione derivatives was prepared with good to excellent yields, and their chemical structures and the purity were proved from spectral and elemental analyses. The work exhibits a feasible solution for promising antibacterial agents, which may offer some insights for subsequent investigations. Thus, we would like to suggest accepting the article after a minor revision.
1. It is better to give the specific experimental effects of these inhibitors
2. The explanation of CADD method is not enough, and the explanation part should improve.
3. Figure 2 should be re-plot, especially the font type and size.
4. Some up-to-date articles can be referred: Nano Energy, 2024, 124: 109498; Nano Energy, 2023, 107: 108132; Energy Conversion and Management, 2022, 269: 116098.
Author Response
Comments and Responses # molecules 3251295 (Reviewer 2)
In this manuscript, A series of new eleven quinazolin-2,4-dione derivatives was prepared with good to excellent yields, and their chemical structures and the purity were proved from spectral and elemental analyses. The work exhibits a feasible solution for promising antibacterial agents, which may offer some insights for subsequent investigations. Thus, we would like to suggest accepting the article after a minor revision.
We are thankful to the reviewers for their valuable comments on our manuscript (Manuscript Number: # molecules 3251295) entitled “Evaluation of quinazolin-2,4-dione Derivatives as Promising Antibacterial Agents: Synthesis, In vitro, In silico ADMET and Molecular Docking Approaches”. The constructive suggestions and comments of the reviewers have greatly helped us in improving our manuscript. We have revised the manuscript in the light of comments of the reviewers, and the corrections have been highlighted in the revised manuscript. The point wise reply to the comments of the learned reviewer 2 is provided below.
- It is better to give the specific experimental effects of these inhibitors
Response
As per the reviewer's suggestion, the necessary corrections are included in the revised manuscript.
- The explanation of CADD method is not enough, and the explanation part should improve.
Response
The necessary corrections are included in the revised manuscript.
- Figure 2 should be re-plot, especially the font type and size.
Response
The necessary corrections are included in the revised manuscript.
- Some up-to-date articles can be referred: Nano Energy, 2024, 124: 109498; Nano Energy, 2023, 107: 108132; Energy Conversion and Management, 2022, 269: 116098.
Response
We understand the importance of the suggested references. We regret to inform you that the suggested references are not suitable for our work.
